# Applying the FAIR4Health Solution to Identify Multimorbidity Patterns and Their Association with Mortality through a Frequent Pattern Growth Association Algorithm

**DOI:** 10.3390/ijerph19042040

**Published:** 2022-02-11

**Authors:** Jonás Carmona-Pírez, Beatriz Poblador-Plou, Antonio Poncel-Falcó, Jessica Rochat, Celia Alvarez-Romero, Alicia Martínez-García, Carmen Angioletti, Marta Almada, Mert Gencturk, A. Anil Sinaci, Jara Eloisa Ternero-Vega, Christophe Gaudet-Blavignac, Christian Lovis, Rosa Liperoti, Elisio Costa, Carlos Luis Parra-Calderón, Aida Moreno-Juste, Antonio Gimeno-Miguel, Alexandra Prados-Torres

**Affiliations:** 1EpiChron Research Group, Aragon Health Sciences Institute (IACS), IIS Aragón, Miguel Servet University Hospital, 50009 Zaragoza, Spain; bpoblador.iacs@aragon.es (B.P.-P.); aponcel@salud.aragon.es (A.P.-F.); aidamorenoj@gmail.com (A.M.-J.); agimenomi.iacs@aragon.es (A.G.-M.); sprados.iacs@aragon.es (A.P.-T.); 2Health Services Research on Chronic Patients Network (REDISSEC), ISCIII, 28029 Madrid, Spain; 3Delicias-Sur Primary Care Health Centre, Aragon Health Service (SALUD), 50009 Zaragoza, Spain; 4Red de Investigación en Cronicidad, Atención Primaria y Promoción de la Salud (RICAPPS), ISCIII, 28029 Madrid, Spain; 5Aragon Health Service (SALUD), 50017 Zaragoza, Spain; 6Division of Medical Information Sciences, Geneva University Hospitals, 1205 Geneva, Switzerland; Jessica.Rochat@hcuge.ch (J.R.); Christophe.Gaudet-Blavignac@hcuge.ch (C.G.-B.); christian.lovis@unige.ch (C.L.); 7Department of Radiology and Medical Informatics, University of Geneva, 1205 Geneva, Switzerland; 8Group of Research and Innovation in Biomedical Informatics, Biomedical Engineering and Health Economy, Institute of Biomedicine of Seville (IBiS), Virgen del Rocío University Hospital/CSIC/University of Seville, 41013 Seville, Spain; celia.alvarez@juntadeandalucia.es (C.A.-R.); alicia.martinez.garcia@juntadeandalucia.es (A.M.-G.); carlos.parra.sspa@juntadeandalucia.es (C.L.P.-C.); 9Department of Geriatric and Orthopedic Sciences, Catholic University of Sacred Heart, 00168 Rome, Italy; carmen.angioletti92@gmail.com (C.A.); rosa.liperoti@policlinicogemelli.it (R.L.); 10Ucibio Requimte, Faculty of Pharmacy, University of Porto, Porto4Ageing, 4050-313 Porto, Portugal; martassalmada@gmail.com (M.A.); emcosta@ff.up.pt (E.C.); 11SRDC Software Research & Development and Consultancy Corporation, Ankara 06800, Turkey; mert@srdc.com.tr (M.G.); anil@srdc.com.tr (A.A.S.); 12Internal Medicine Department, Virgen del Rocío University Hospital, 41013 Seville, Spain; jaraeloisa@hotmail.com

**Keywords:** FAIR principles, multimorbidity, mortality, research data management, pathfinder case study, privacy-preserving distributed data mining

## Abstract

The current availability of electronic health records represents an excellent research opportunity on multimorbidity, one of the most relevant public health problems nowadays. However, it also poses a methodological challenge due to the current lack of tools to access, harmonize and reuse research datasets. In FAIR4Health, a European Horizon 2020 project, a workflow to implement the FAIR (findability, accessibility, interoperability and reusability) principles on health datasets was developed, as well as two tools aimed at facilitating the transformation of raw datasets into FAIR ones and the preservation of data privacy. As part of this project, we conducted a multicentric retrospective observational study to apply the aforementioned FAIR implementation workflow and tools to five European health datasets for research on multimorbidity. We applied a federated frequent pattern growth association algorithm to identify the most frequent combinations of chronic diseases and their association with mortality risk. We identified several multimorbidity patterns clinically plausible and consistent with the bibliography, some of which were strongly associated with mortality. Our results show the usefulness of the solution developed in FAIR4Health to overcome the difficulties in data management and highlight the importance of implementing a FAIR data policy to accelerate responsible health research.

## 1. Introduction

Chronic conditions are responsible for most health problems in older people [1], in which multimorbidity, or coexistence of multiple chronic diseases, is the norm rather than the exception. There is growing scientific evidence that chronic conditions tend to cluster into specific non-random disease patterns, commonly referred to as multimorbidity patterns [2]. Nonetheless, most health systems and clinical practice guidelines are still designed to respond to specific diseases independently. Consequently, the needs of people with multimorbidity are often not adequately met by health care services and health professionals, resulting in avoidable negative effects on health and healthcare costs for patients and health systems, respectively [3]. Multimorbidity is commonly followed by polypharmacy or the prescription and use of multiple (five or more) medications by the patient. Multimorbidity and polypharmacy are associated with increased mortality and cognitive impairment and decreased quality of life and functional ability [3,4,5].

Different initiatives have tried to face the challenge of managing multimorbidity in clinical practice in recent years. Some of them are conceptual and propose new models of care based on the comprehensive assessment of the patient and shared decision making, such as the Ariadne principles [6] or the Integrated Multimorbidity Care Model [7], which have even been tested in clinical trials and real-life situations. Another feasible and less expensive option for obtaining scientific evidence on multimorbidity is to carry out studies based on real-world data. The current availability of health data, such as those contained in patients’ electronic health records (EHR) represents an excellent opportunity for health research. However, it also presents some limitations, such as difficulties in performing cross-regional studies due to the lack of interoperability among datasets, problems related to data access and privacy and other challenges associated with the current lack of tools to harmonize and integrate health datasets.

In this context, the FAIR4Health project was born as a Horizon 2020 project aimed at facilitating and encouraging the European health research community to share, reuse and integrate publicly-funded research datasets [8] based on the FAIR (findability, accessibility, interoperability and reusability) principles that serve to guide scientific data management and stewardship [9]. A “FAIRification Workflow” to apply the FAIR principles to EHR and other health research data sources was designed and implemented [10] by adapting the GO FAIR process to health data’s legal, ethical and technical requirements. In addition, a common data model was defined to allow federated data analysis through the integration of datasets from various health research organizations. Furthermore, different software tools were developed in FAIR4Health to implement the FAIRification Workflow. Among them, the Data Curation Tool [11,12] served to integrate raw health datasets by transforming them into FAIR datasets, whereas the Data Privacy Tool [13] preserved data privacy through data de-identification and anonymization methods. On top of these, the FAIR4Health Platform was developed to provide a set of services for the researchers in a user-friendly interface, aimed at allowing the application of federated machine learning algorithms on FAIR datasets.

To demonstrate the potential usefulness for health research of this FAIRification strategy, two pathfinder case studies were performed by using federated machine learning algorithms implemented upon the FAIR4Health Platform. In this paper, we present the results obtained in the FAIR4Health pathfinder case study aimed at identifying multimorbidity patterns in older adults from different healthcare settings and analyzing their impact on mortality through a frequent pattern growth association algorithm.

## 2. Materials and Methods

We conducted a multicentric retrospective observational study that included five European cohorts from different healthcare settings (i.e., hospital, primary care, nursing homes) and health research organizations: Université de Genève (UNIGE, Switzerland), Università Cattolica del Sacro Cuore (UCSC, Italy), University of Porto (UP, Portugal), Instituto Aragonés de Ciencias de la Salud (IACS, Spain) and Andalusian Health Service (SAS, Spain). Each of these organizations provided a database from a publicly funded research project for the purposes of the study.

UNIGE provided anonymized healthcare data from the EHR of the University Hospitals of Geneva, the largest one in Switzerland, and met the needs of around 0.5 M residents (*n* = 244). UCSC provided health data from two research studies carried out within the SHELTER project [14,15] that aimed to implement a tool to assess and collect data about nursing home residents (*n* = 331). UP provided a health research dataset based on the FRAILSURVEY study [16], which aimed to test the reliability of the FRAILSURVEY phone app for self-assessment of frailty in older adults (*n* = 861). IACS provided a health research dataset based on the EpiChron Cohort [17], which investigates the clinical epidemiology and health outcomes associated with chronic diseases and multimorbidity in the Spanish region of Aragon (*n* = 3786). SAS provided health care data from the EHR of Virgen del Rocío University Hospital of Seville, one of the biggest hospitals in Spain, and covering a population of more than 0.5 M inhabitants (*n* = 5812).

The study population included patients over 65 years of age with multimorbidity (i.e., at least two chronic diseases). During the study, researchers from each institution carried out a secondary use of retrospectively collected data using federated machine learning algorithms. Ethical approval for this study was obtained in all countries based on local regulations (UNIGE, 2020-02683; UCSC, 1066/20-12/05/2020; UP, PARECER A-13/2020; SAS, 1269-M1-20; and IACS, 1269-M1-20).

### 2.1. Study Variables

UNIGE, UCSC, IACS and SAS shared a similar data structure and information on the same disease variables, allowing to analyze the four datasets together and, in this way, identifying multimorbidity patterns. We studied the following variables for each patient at cohort entry: age, gender, nationality, smoking status, institutionalization, polypharmacy (i.e., use of ≥5 drugs), number of prescribed drugs for those with polypharmacy and the presence of 47 chronic baseline conditions registered in patients’ EHR. The selection of the 47 conditions analyzed was based on clinical consensus. Additionally, SAS and IACS datasets were used to analyze the impact of multimorbidity on mortality at six months due to their shared structure regarding this outcome.

The UP dataset included the following variables: age, gender, nationality, memory complaints, vision/hearing difficulties, unintentional weight loss, feeling depressed lately, feeling anxious lately, Groningen Frailty Index Frailty Score and domiciliary care. Therefore, it was analyzed independently to identify multimorbidity patterns.

### 2.2. FAIRification Workflow and Tools Developed

FAIR4Health extended the FAIRification process adopted by the GO FAIR initiative [18] for the health domain, considering specific technical, ethical and legal requirements. As a result, a FAIRification Workflow [10] consisting of the following 10 steps was introduced: (1) raw data analysis, (2) data curation & validation, (3) data de-identification & anonymization, (4) semantic modeling, (5) making data linkable, (6) license attribution, (7) data versioning, (8) indexing, (9) metadata aggregation and (10) publishing.

In order to achieve the objectives of the FAIRification Workflow, specific software tools were developed and utilized to enable data managers to make their raw health research data FAIR. The Data Curation Tool [11,12] represented the entry point to the FAIRification workflow. Its main goal is to annotate clinical datasets with medical terminologies and define mappings to the FAIR4Health Common Data Model [19], implemented following the HL7 FHIR profiling approach [20]. This tool wrote data into a HL7 FHIR repository instance by processing the raw source data according to the mapping rules defined by the data manager. It has been proven to be an effective tool that meets the fundamental requirements of raw data analysis, curation and validation [21]. On the other hand, the Data Privacy Tool [13] was responsible for handling the privacy challenges on sensitive health data by applying several data de-identification and anonymization techniques. Once the curation process was finished, data managers used this tool to de-identify data before making it available to other systems/components as FAIR data.

The high-performance secure health data repository onFHIR.io [22], which is totally compliant with the HL7 FHIR specifications, was utilized as the HL7 FHIR Repository. This repository stored and maintained the data made FAIR according to the FAIR4Health Common Data Model, satisfying the objectives of the FAIRification workflow, such as licensing, versioning, indexing and publishing.

In order to show the potential impact of the FAIRification strategy, a Privacy-Preserving Distributed Data Mining (PPDDM) architecture was implemented to build machine learning models in a federated way. The architecture consisted of two main components: the aforementioned FAIR4Health Platform and the FAIR4Health Agents, which were a suite of software applications installed locally at each participating sites’ own system that not only provide the FAIRification tools, but also host the PPDDM Agent responsible for running machine learning algorithms on FAIR data and exchanging the results with the FAIR4Health Platform. Thus, no health data were shared among participating sites or with the FAIR4Health Platform.

The FAIR4Health Platform, on the other hand, provided a set of services with an elaborate Graphical User Interface (GUI) on top in charge of interacting with the agents and orchestrating the whole process.

### 2.3. Analysis

We applied a federated frequent pattern growth association (FP-Growth) algorithm [23], used for mining association rules, to identify the most frequent patterns among the set of variables studied. FP-Growth is an efficient, scalable and fast algorithm implemented by Han et al. [23] for mining frequent patterns, especially when the size of data and/or the number of variables are large. We implemented the FP-Growth algorithm in a federated manner in line with the PPDDM objectives so that no real data were shared between the participants. Given a dataset in a number of agents, the association rules were identified in two steps.

In the first step, each PPDDM agent calculated the item frequencies on their own data through the construction of a FP-tree and sent the results to the PPDDM Manager in the FAIR4Health Platform. The PPDDM Manager merged the results of all agents, found frequencies at the global level, and removed the ones below a minimum threshold value referred to as support that ranged from 0.0 to 1.0. For a disease, support could also be considered as its prevalence. For example, if an item appeared in 5 out of 10 records, it had a support of 5/10 = 0.5. We considered 0.3 as the default minimum support value. The lower the minimum support, the more variables were included in the next step.

In the second step, the PPDDM Manager sent the global itemset to each agent and asked agents to find association rules containing items from this itemset. For each item, the conditional FP-tree was built, and the association rules were generated. Then, the confidence (i.e., how often an association was observed in the dataset) was calculated for each rule. For instance, if itemset X appeared five times, and X and Y appeared together three times, the confidence for the association rule X ≥ Y was 3/5 = 0.6.

Once association rules were generated, they were sent to the PPDDM Manager. Similar to the process in the first step, the PPDDM Manager combined the association rules retrieved from all the participating agents, calculated the global confidence values, and eliminated the ones having a confidence lower than the minimum value allowed, which was 0.8 by default. As a result, the remaining association rules constituted the frequent patterns discovered in the datasets. The FAIR4Health Platform presented the association rules in an “Antecedent ≥ Consequent” format, as shown in Figure 1.

The “antecedent” column represented the left-hand side of an association rule, while the “consequent” column represented the right-hand side. The “confidence” column defined the probability that a patient had the “consequent” conditions given that he/she already had the “antecedent” ones. For example, the association rule shown in Figure 1 indicates that a male patient with heart failure, hyperlipidemia, hypertension and polypharmacy presents a likelihood of suffering diabetes mellitus of 65.6% (0.656). For an association rule A ≥ C, confidence was calculated as the ratio between patients having A and C, and patients having A.

The “lift” or correlation, on the other hand, indicates whether having the “antecedent” conditions (A) actually increases the probability to have the “consequent” conditions (C). It was calculated as the ratio between the confidence of A ≥ C and support of C, and could be interpreted as a measure of the strength of association. In cases where “A” actually led to “C” (i.e., positive correlation), the lift value was greater than 1. In other words, the greater the lift value, the more likely the patient to have “C” given that he/she already has “A”. However, if the confidence was high but the value of lift was less than 1 (i.e., negative correlation), then we concluded that having “A” for a patient did not increase the likelihood of presenting “C”. In the example in Figure 1, the lift value of 1.663 indicates a strong association.

Following this methodology, different models were created adjusting the minimum support or frequency a variable should have to be included (i.e., the prevalence in the case of a disease), and the minimum confidence or frequency an association rule should have to be presented. The lower the minimum support and minimum confidence were, the higher the number of diseases and associations included was. As a result, hundreds of rules for each model can be obtained. However, only those association rules with positive correlation and highest clinical relevance and confidence will be presented in the results section in order to show the potentiality of the tools developed.

## 3. Results

The demographic characteristics of the 11,034 individuals included in the datasets used in the study are summarized in Table 1. The mean age of the population was 82.1 years and women represented almost 51% of the individuals studied.

### 3.1. Identification of Multimorbidity Patterns

The most frequently identified chronic conditions were cardio-metabolic (i.e., diabetes mellitus, hyperlipidemia, hypertension and obesity), cardiovascular (i.e., heart failure and chronic kidney disease) and mental (i.e., depression and anxiety). The most relevant multimorbidity patterns found, based on combinations of the parameters used in the models, are presented in Table 2.

The pattern with the highest strength of association (2.796) consisted in the presence of atrial fibrillation, chronic anemia, chronic kidney disease, coronary heart disease, hypertension and polypharmacy, which also resulted in the appearance of heart failure (probability of confidence, 0.86 out of 1). We also found a multimorbidity pattern consisting of a polymedicated patient with atrial fibrillation, chronic anemia, chronic kidney disease, coronary heart disease, diabetes mellitus, heart failure and hyperlipidemia, who also develops hypertension (probability of confidence, 1; lift, 1.33).

As explained in the methodology, UP dataset was analyzed independently due to its different data structure. In this case, the pattern with the highest lift (2.52) consisted of a male patient, aged 70–80 years, feeling down or depressed and nervous or anxious lately and with memory complaints and vision difficulties, who also develops hearing difficulties (confidence, 0.91). We found some patterns with perfect confidence, such as one consisting of a male patient, aged 80 and older, feeling down or depressed and nervous or anxious lately and with hearing and vision difficulties, who was then polymedicated (lift, 1.65).

### 3.2. Impact of Multimorbidity Patterns on Mortality

The multimorbidity pattern with the highest positive correlation with mortality consisted of chronic anemia, chronic kidney disease, coronary heart disease, diabetes mellitus and heart failure, which was then associated with mortality with a probability of confidence of 0.58 out of 1 and a lift of 1.96 (Table 3).

In patients with polypharmacy in its antecedents, the highest correlation with mortality was presented in those with chronic anemia, chronic kidney disease, coronary heart disease, diabetes mellitus, heart failure and hyperlipidemia (probability of confidence, 0.54 out of 1, correlation, 1.82).

## 4. Discussion

In this study, we tested the usefulness of the FAIR4Health solution to apply the FAIR principles in health research by developing a pathfinder case study aimed at identifying multimorbidity patterns and their impact on mortality based on a federated data analysis on five datasets from different European health research organizations using PPDDM methodologies and a frequent pattern growth algorithm.

The objectives proposed by the project’s clinical researchers were satisfactorily addressed in the pathfinder case study. Cardiometabolic and mental health patterns were identified among the most frequent and relevant ones in our study, a result consistent with previous studies [2,24]. The systematic review by Busija et al. in 2019 [24] concluded that the only replicable and clinically meaningful multimorbidity profiles are the cardiometabolic and mental health; a previous systematic review by Prados-Torres et al. in 2014 [2], described three main multimorbidity patterns, cardiometabolic, mental health and musculoskeletal. These results largely coincide with our findings and support the existence of the multimorbidity patterns identified, besides a strong association between multimorbidity with mortality was described, demonstrating the potentiality of our FAIRification strategy on health research and, hopefully, on patients’ health outcomes.

Another potentiality and novelty of our study is that we can analyze the antecedents and consequences of the patterns detected. This approach can help to identify key associations that lead to specific consequences, analyzing the clinical impact of the patterns using the diseases as the study unit. From a clinical point of view, this is relevant as it can help to develop preventive actions based on the disease associations and the frequency of their appearance. However, we must be careful about the clinical results obtained in this study that should be interpreted with caution.

The FAIRification workflow and tools developed allowed us to analyze heterogeneous datasets and to increase the variability of studied datasets (i.e., more detailed clinical, demographic, environmental and social information) compared with studies not applying FAIR principles and always in a secure way.

However, we had to face some challenges regarding data collection, which, at the same time, helped us to create cross-cutting solutions in the process. First, the data extraction of EHR and other health research data sources had to be aligned with the FAIR4Health Common Data Model and which required relevant efforts. Each participating organization in the data extraction involved experts in their source data model in tackling these problems, which improved the communication between different specialists from different areas, an essential element for research dynamic. In some cases, the application of natural language processing (NLP) techniques to handle the information in free text fields was required, developing human–machine interaction skills fundamental for this project and in future ones. Finally, to deal with the differences between the types of raw data sources (e.g., research and clinical-administrative datasets), we analyzed each source raw dataset in-depth in a collaborative effort between clinical and technical researchers. All this led us to reach the precise configuration to apply FAIR principles within the FAIR4Health solution, making all raw data FAIR and then generating PPDDM models using all data sources. A pathfinder project like this probably can help to build multidisciplinary teams essential in health research to face new challenges. The application of FAIR principles and the tools developed in this project have great potentiality in different health research contexts; they can be applied to different types of datasets and can help to answer different research questions, which can help us to guide scientific data management and drive scientific discovery to a new paradigm.

Regarding the limitations of the study, some of them were related to the association patterns obtained. It would be possible to generate more efficient association rules if we could better adjust the mortality variable distribution in our datasets, including a larger number of patients and from other regions, and, in this way, control the risk of bias. We also have to consider the computational limitations of the frequent pattern growth association algorithm applied in this study. When clinical researchers decreased the minimum confidence and support values to include diseases with low prevalence, the number of combinations increased, and the model could not get the results. To address this challenge, other types of associative methods, such as factor, cluster and network analysis [25,26,27] could be explored and implemented in future works.

## 5. Conclusions

The use of the FAIR4Health solution enabled us to identify multimorbidity patterns and their association with mortality in older adults using complex and heterogeneous FAIR databases from different European countries. Our results show the potential of implementing a FAIR data policy in health research and support the usefulness of the FAIR4Health solution, encouraging the scientific community to use the tools developed to test and validate their performance in different research contexts.

## Figures and Tables

**Figure 1 ijerph-19-02040-f001:**
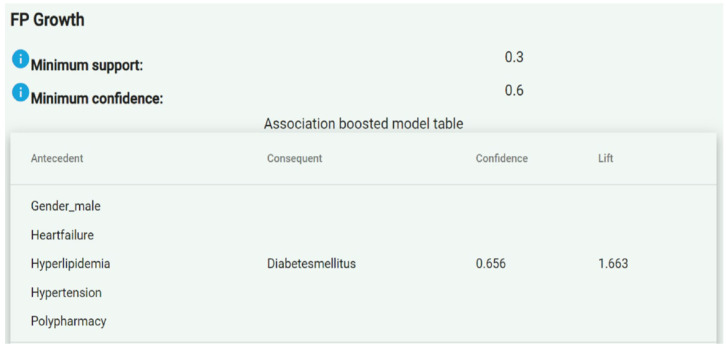
Example of presentation of the association rules in the FAIR4Health Platform.

**Table 1 ijerph-19-02040-t001:** Demographic characteristics of the populations from the five agents included in the study.

Institutions	Population (*n*, %)	Age (Mean)	Sex, Women (%)
Université de Genève	244 (2.2)	81.8	47.1
Università Cattolica del Sacro Cuore	331 (3.0)	95.5	71.6
University of Porto	861 (7.8)	76.6	57.5
Instituto Aragonés de Ciencias de la Salud	3786 (34.3)	82.1	49.9
Andalusian Health Service	5812 (52.7)	82.2	49.4
Total	11,034 (100)	82.1	50.8

**Table 2 ijerph-19-02040-t002:** Multimorbidity patterns found in the study population based on the selected combinations of parameters used in the models.

Parameters Used		Generated Patterns				Institutions Providing Datasets in Each Model
Minimum Support	Minimum Confidence	Antecedent (A)	Consequent (C)	Confidence	Correlation (Lift)	
0.2	0.5	Atrial fibrillation Chronic anemia Chronic kidney disease Coronary heart disease Hypertension Polypharmacy	Heart failure	0.86	2.80	UNIGE, UCSC, IACS, and SAS
0.2	0.5	Atrial fibrillation Chronic anemia Chronic kidney disease Coronary heart disease Diabetes Mellitus Heart failure Hyperlipidemia Polypharmacy	Hypertension	1.00	1.33	UNIGE, UCSC, IACS, and SAS
0.3	0.5	Gender male Age 70–80 Feeling down or depressed lately Feeling nervous or anxious lately Memory complaints Vision difficulties	Hearing difficulties	0.909	2.52	UP
0.3	0.5	Gender male Age 80 and older Feeling down or depressed lately Feeling nervous or anxious lately Hearing difficulties Memory complaints Vision difficulties	Polymedicated	1.00	1.65	UP

UNIGE: Université de Genève; UCSC: Università Cattolica del Sacro Cuore; IACS: Instituto Aragonés de Ciencias de la Salud; SAS: Andalusian Health Service; UP: University of Porto.

**Table 3 ijerph-19-02040-t003:** Impact of multimorbidity patterns on mortality based on the selected combinations of the parameters used in the models.

Parameters Used		Generated Patterns			
Minimum Support	Minimum Confidence	Antecedent (A)	Consequent (C)	Confidence	Correlation (Lift)
0.2	0.8	Chronic anemia Chronic kidney disease Coronary heart disease Diabetes mellitus Heart failure	Mortality	0.58	1.96
0.2	0.8	Chronic anemia Chronic kidney disease Coronary heart disease Diabetes mellitus Heart failure Hyperlipidemia Hypertension	Mortality	0.55	1.85
0.2	0.8	Chronic anemia Chronic kidney disease Coronary heart disease Diabetes mellitus Heart failure Hyperlipidemia Polypharmacy	Mortality	0.54	1.82

## Data Availability

Not applicable.

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
