# Peer review of "Applying the FAIR4Health Solution to Identify Multimorbidity Patterns and Their Association with Mortality through a Frequent Pattern Growth Association Algorithm"

_ijerph, 2022, doi:10.3390/ijerph19042040_

Round 1

Reviewer 1 Report

This is an interesting multicenter study on the feasibility of an harmonized machine learning analysis on the association between multimorbidity and mortality.Some issues should be clarified before publication:- the study/population included seem all to be research sample studies. Are they all prospective clinical studies? Lack of study based on administrative health records should be mentioned as the possibility of applying the methods to such databases.- has follow-up for mortality been considered in these studies (and in your analysis)?-the authors applied a FP-growth algorithm. Have you considered other ML algorithms as sensitivity analysis?

Author Response

This is an interesting multicenter study on the feasibility of an harmonized machine learning analysis on the association between multimorbidity and mortality. Some issues should be clarified before publication:

- the study/population included seem all to be research sample studies. Are they all prospective clinical studies? Lack of study based on administrative health records should be mentioned as the possibility of applying the methods to such databases.

Thanks for your comment.

Regarding your appreciation,“the study/population included seem all to be research sample studies” , we have performed a study that included a population from both kinds of samples: research and clinical-administrative data.

Regarding the first question, some of the studies originally were retrospective and some other prospective. However, we performed a transversal analysis in our study, and this classification is not relevant.

The FP algorithm uses the antecedent and consequent logical terminology as the result of its analysis. This terminology in a transversal analysis implies association but does not imply casualty or temporal analysis. Except for the variable mortality (at six months) in which we considered the longitudinal impact.

We have included some sentences in the discussion to answer these comments and highlight the potentiality of the tools developed in this study that could be applied to different kinds of health datasets.

Please see lines 339, 344-347, page 9.

- has follow-up for mortality been considered in these studies (and in your analysis)?

Thanks for your comment. Mortality has a follow-up at six months in our analysis. SAS and IACS datasets were used for mortality analysis, and they do not have a specific follow-up. We have added a sentence to detail this question.

Please see lines 148, page 3.

-the authors applied a FP-growth algorithm. Have you considered other ML algorithms as sensitivity analysis?

In the literature, there are different types of approaches to achieve association rule learning and there are number of algorithms that have been developed by researchers in each approach. For instance, k-optimal classification rule discovery algorithm is built on k-optimal pattern discovery approach which finds k patterns in the data, while Apriori, Eclat and FP-Growth are examples of algorithms built on frequent pattern discovery approach which finds all patterns in the data that are sufficiently frequent. In this study, our aim is to find frequent item sets in a federated way without violating the privacy of participating organizations, hence we focused on the frequent pattern discovery approach. We considered all the algorithms existing in this approach, but we observed that FP-Growth is developed on top of its predecessors, Apriori and Eclat, and it has been proven to be better algorithm than the others. Therefore, in our study, we only implemented the federated version of FP-Growth.

Reviewer 2 Report

This is an excellent research that tested the usefulness of the FAIR4Health solution to apply the FAIR  principles in health research by developing PPDDM methodologies and a frequent pattern growth algorithm. 

Although the main purpose of the present study was to develop new analytical techniques and apply usefulness, it is necessary to additionally describe in the DISCUSSION section: what the multimorbidity patterns shown as a result of this study has the similarities and differences from previous studies, and what practical implications it has for medical practice. If there is  this part is revised and improved, it is expected that the quality of the research will be improved as an excellent study that is meaningful both methodologically and practically.

Author Response

This is an excellent research that tested the usefulness of the FAIR4Health solution to apply the FAIR  principles in health research by developing PPDDM methodologies and a frequent pattern growth algorithm. 

Although the main purpose of the present study was to develop new analytical techniques and apply usefulness, it is necessary to additionally describe in the DISCUSSION section: what the multimorbidity patterns shown as a result of this study has the similarities and differences from previous studies, and what practical implications it has for medical practice. If there is  this part is revised and improved, it is expected that the quality of the research will be improved as an excellent study that is meaningful both methodologically and practically.

Thanks for your comment. We have improved the discussion as suggested. Please see lines 308-316 and 318-324, page 9.

Reviewer 3 Report

This is an interesting article covering rule mining in the health context in a unique manner.  The paper is well written. Also, the approach of having association rule mining conducted at various nodes and then combining these centrally sounds like a very interesting approach, involving human-system interaction. 

Just a few minor editing suggestions.

First, in line 305, may replace "what" with "and which" to address an expression issue.

Also, for this paragraph in the discussion (lines 302-317), it appears the focus is to highlight the unique steps undertaken.  I think this part better fits in a methodology section than a discussion section. The discussion section, rather, may reflect the extent the adopted methodology is unique compared to the existing literature and the potentials of these steps being applied in other health context reserach.    

There may also be some indications of future works. 

Overall, this article is acceptable. 

Author Response

This is an interesting article covering rule mining in the health context in a unique manner.  The paper is well written. Also, the approach of having association rule mining conducted at various nodes and then combining these centrally sounds like a very interesting approach, involving human-system interaction. 

Just a few minor editing suggestions.

First, in line 305, may replace "what" with "and which" to address an expression issue.

Thanks for your comment. We have changed it as suggested. Please see line 332, page 9.

Also, for this paragraph in the discussion (lines 302-317), it appears the focus is to highlight the unique steps undertaken.  I think this part better fits in a methodology section than a discussion section. The discussion section, rather, may reflect the extent the adopted methodology is unique compared to the existing literature and the potentials of these steps being applied in other health context research.  There may also be some indications of future works. Overall, this article is acceptable. 

Thanks for your comment. We have changed the paragraph focusing on the potentials of the steps applied.

Please see lines 332-362, pages 9-10.